# Nurses' perspectives on using mobile health applications in southeastern Iran: Awareness, attitude, and obstacles

**Jahanpour Alipour**[1], **Yousef Mehdipour**[2], **Somayyeh Zakerabasali**[3], **Afsaneh Karimi**[4]*

1 Health Human Resources Research Center, School of Health Management and Information Sciences, Shiraz University of Medical Sciences, Shiraz, Iran, 2 Department of Health Information Technology, School of Paramedical, Torbat Heydarieh University of Medical Sciences, Torbat Heydarieh, Iran, 3 Clinical Education Research Center, Health Human Resources Research Center, Department of Health Information Management, School of Health Management and Information Sciences, Shiraz University of Medical Sciences, Shiraz, Iran, 4 Pregnancy Health Research Center, Zahedan University of Medical Sciences, Zahedan, Iran

* afsanehkarimi2014@gmail.com

## Abstract

### Introduction

Nurses and patients can now ensure access to qualified healthcare using the new opportunities of mobile health (mHealth) applications (or apps). To use its potential effectively, understanding nurses' use of this technology is crucial. Here, we examined the awareness, attitudes, and obstacles to using mHealth apps among nurses.

### Methods

We applied a descriptive-analytical cross-sectional study from 08/04/2023 to 05/10/2023. Cochran's formula estimated the sample size of 267 nurses. The researchers used a researcher-made questionnaire for data collection. We used descriptive (mean, standard deviation, frequency, and percentage) and analytical (Pearson and Spearman correlation) statistics for data analysis.

### Results

Most nurses use a smartphone (86%), have an Android operating system installed (82%), and believe that mHealth is useful for nursing (85%), but do not use it for patient care (70.8%). The mean score for awareness, attitude, and obstacles regarding mHealth were $3.74 \pm 0.657$, $3.49 \pm 0.513$, and $3.50 \pm 0.597$ respectively. There was a significant positive correlation between the nurses' awareness and attitude ($r = 0.289$, $P < 0.05$) and nurses' awareness and obstacles to using mHealth ($r = 0.171$, $P < 0.05$), but a significant negative correlation between nurses' attitude and obstacles ($r = -0.031$, $P < 0.05$).

### Conclusion

Despite nurses' relatively favorable awareness and positive attitude towards the use of mHealth apps, most of them use these technologies for purposes other than patient care.

**Data availability statement:** All relevant data are within the paper and its Supporting Information files.

**Funding:** This study was supported by the Vice Chancellor for Research and Technology of Zahedan University of Medical Sciences (No: 10494).

Nurses considered patients' lack of digital health literacy (DHL) and reluctance to use these technologies as the main obstacles to using mHealth apps. Improving the DHL of users (nurses and especially patients) and providing them with free access to mHealth apps is essential. Ensuring security and making the applications easy to use, as well as educating users, are also important factors. Furthermore, promoting nurses' understanding of the benefits of mHealth and increasing their willingness to use these technologies is crucial.

## Introduction

In the healthcare industry, health information technologies are used to provide health care and promote public health [1,2]. In low- and middle-income countries, the number of mobile phone users has escalated in the last decade and has provided the potential to improve the efficiency of healthcare [3,4]. Currently, almost 95% of people across the world have access to the mobile network [3]. In 2022, approximately 67 million people in Iran are estimated to be smartphone users, representing a substantial portion of the country's total population of 88.55 million [5], which indicates a favorable context for the use of mobile health (mHealth) technologies. In Iran, mHealth technology has been used in the field of managing heart [6], lung [7], diabetes [8], depression [9], and dialysis [10].

Hence, mHealth is a promising, creative, and effective technological solution for healthcare workers to provide various healthcare services for patients, especially those with chronic diseases [11,12]. mHealth refers to the use of mobile and wireless technologies such as mobile phones, patient monitoring devices, personal digital assistants, wireless sensors, and software applications used to achieve healthcare goals [3,13]. This technology can be used in the clinical field in all continuum of care (prevention, diagnosis, treatment, follow-up, and patient education and monitoring) [14]. In addition, the use of these technologies by patients has a significant role in improving the self-care of patients with chronic diseases such as cancer, diabetes mellitus, and cardiovascular diseases [4,15].

In the last decade, the number of mobile applications used by healthcare professionals has increased exponentially. However, the lack of knowledge for professional use, training requirements, and understanding of the quality of these apps among healthcare professionals, including nurses, have been mentioned as the reasons for resistance to the use of this technology [16]. Nurses are the largest group of frontline workers in the healthcare industry in most countries, they play a crucial role in providing direct clinical care, counseling, follow-up of patients' treatment, education to patients or their families, promoting public health, and empowering patients through improving the self-care approach [16,17].

According to the estimate made by 2030, the healthcare industry in the world will face a shortage of 15 million workers. This deficiency will be more noticeable in countries with higher-than-average income, including Iran [18]. The nursing staff is one of the main groups providing healthcare and the efficiency of the health system is directly affected by the performance of this group of staff. The shortage of nurses is a universal challenge. Iran is also facing the challenge of a shortage of nurses, which can lead to negative consequences such as job dissatisfaction, an increase in professional errors, a decrease in the quality of healthcare, and even a surge in patient mortality [19]. Nurses use mHealth for various purposes, such as supporting their daily work activities, documenting patients' records [17], accessing evidence-based knowledge, evidence-based decision-making [20], promoting their skills, and patients' educating [21]. Accordingly, mHealth can facilitate the monitoring

of symptoms and the general health of the patient [22], the discovery of unwanted events, early intervention [23], and finally improving the results of patient care and reducing health care costs [24].

Because of the sensitivity of patient clinical data and strict rules for the use of mHealth tools in the healthcare industry, nurses need to have complete knowledge about how to work with mHealth applications, the expected results, and how to evaluate the accuracy of the information created. Due to the increasing number of patients with chronic diseases, the essential role of nursing staff in managing the care of these patients is more and more apparent. By empowering patients to use mHealth applications to manage chronic diseases, public health can be improved, and nurses have a vital role in this field [25]. Despite the existence of more than 350,000 mHealth apps on top app stores universally [26,27], the use of these apps by nurses is not common [27,28]. In Spain, Mayer et al. [16] revealed that more than half of nurses (51.6%) did not use mHealth professional apps.

Considering the direct and long-term relationship of nursing staff with patients and the variety of care performed by nurses for patients (preventive, therapeutic, and educational care and its documentation), the positive understanding and willingness of this group to use mHealth technology in the patients' care can have a significant effect in promoting self-care of patients and improve the quality of care and increase the accessibility of services for patients even in remote areas. Considering this importance, this study aimed to evaluate the nurses' awareness, attitude, and obstacles ahead toward using mHealth apps in teaching hospitals affiliated with Zahedan University of Medical Sciences.

## Methods

### Study design and setting

In this descriptive-analytical cross-sectional study, conducted from 08/04/2023 to 05/10/2023, we applied our research in five teaching hospitals affiliated with Zahedan University of Medical Sciences, including Ali-Ibne-Abitaleb (a general hospital), Khatam-Ol-Anbia (Trauma center), Alzahra (specialized ophthalmology hospital), Baharan (Specialized psychiatric hospital(, and Buali (specialized infections disease hospital). The Ethics Committee of the Deputy of Research and Technology of Zahedan University of Medical Sciences approved the study (No: IR.ZAUMS.REC.1401.133). Before completing the questionnaire, all participants provided their consent both orally and writing.

### Study population and sample

Our study population consisted of 867 nursing staff members from five teaching hospitals. Using Cochran's formula ($n = pqz^2/d^2$) with a confidence level of 95% and a 5% margin of error 267 nurses was necessary. We employed stratified sampling to select the nursing samples in the hospitals. To account for potential incomplete questionnaires, the sample size was increased to 280 participants. The inclusion criteria for the study were as follows: 1. Nurses who expressed a willingness to participate in the research 2. Nurses who had experience using mHealth, even if only through sending health-related messages or engaging in phone conversations.

### Data collection tool

Data was collected using a paper-based self-administered questionnaire developed by the research team based on previous literatures [4,29–32]. The questionnaire consisted of eight parts. The first part contained four demographic characteristics of participants: age, sex,

work experience, and education level. The second part included seven questions about mHealth applications: mobile device type, installed operating system on the device, applications in use, frequency of using applications per day, an effective source/individual in choosing an application, and the usefulness of mHealth applications from the nurses' perspectives. The third part focused on the nurses' attitude regarding the importance of mHealth apps' capabilities with a five-point Likert scale (from 1: very low to 5: very high) for rating questions (n = 8). The fourth part contained a question about nurses' satisfaction with their awareness regarding mHealth with a 10-point scale (ranging from 1: not at all satisfied to 10: extremely satisfied). The fifth section comprised 15 questions assessed participants' awareness regarding mHealth with a five-point Likert scale (ranging from 1: completely disagree to 5: completely agree) for rating questions. The sixth part included a yes/no question about whether participants had education on mHealth. The seventh section examined nurses' willingness to participate in mHealth training courses using five questions with a five-scale Likert scale (ranging from 1: very low to 5: very high). The final section contained 14 questions about the obstacles to using mHealth from the nurses' perspectives with a five-point Likert scale (ranging from 1: very low to 5: very high) for rating questions.

The questionnaire's validity and reliability were confirmed. The questionnaire was validated by 10 experts, including six health information management experts and four medical informatics experts with PhD degrees. Its face and content validity were calculated based on the impact score indicator (IPS), content validity index (CVI), and content validity ratio (CVR) [33]. Internal consistency was measured using Cronbach's alpha index and reliability over time was assessed using the Intraclass Correlation Coefficient Index (ICC) [34], all of which were above 80% and approved.

## Statistical analysis

Data from the questionnaire were manually entered into SPSS software by the researchers. Data quality control and cleaning were done by researchers using basic descriptive statistics to identify and resolve potential errors, shortcomings, and issues before further data analysis. Descriptive (mean, standard deviation, frequency, and percentage) and analytic (spearman and Pearson correlations) statistics analysis were conducted using the Statistical Package for Social Sciences (SPSS) version 22.0™ (IBM Corporation, Los Angeles, CA, USA). Mean scores of the factors were used to evaluate the desirability of awareness and attitude of the nurses regarding mHealth apps. The following scale was used for interpretation. A mean score of 3.75 or more out of 5 indicated that a factor was considered desirable; a mean score of 3 to less than 3.75 out of 5 suggested that a factor was considered relatively desirable, and a mean score of 1.5 to less than 3 was deemed undesirable. The obtained r values were interpreted according to Akoglu [35] for the evaluated factors. A p-value < 0.05 was considered significant.

## Results

A total of 280 questionnaires were distributed, out of which 271 were completed and returned, resulting in a response rate of 96.78%. Four questionnaires were excluded due to many defects in filling the essential fields; thus, the data from 267 questionnaires was entered into the final analysis.

According to results, the majority of the participants were female (73%) and in the age group of 31-40 years (59.6%), and had a bachelor's degree (91.4%). The most frequent work experience was 9-16 years (43.8%) (Table 1).

**Table 1. Demographic characteristics (n = 267).**

| Category | Subcategory | Frequency | |
|---|---|---|---|
| | | Number | Percent |
| Gender | Female | 195 | 73 |
| | Male | 72 | 27 |
| Age (year) | 24-31 | 51 | 19.1 |
| | 32-39 | 159 | 59.6 |
| | 40-48 | 57 | 21.3 |
| Marital status | Single | 181 | 67.8 |
| | Married | 86 | 32.2 |
| Education Degree | Associate's Degree | 3 | 1.1 |
| | Bachelor's | 244 | 91.4 |
| | Master's | 20 | 7.5 |
| Work experience (year) | ≤ 8 | 86 | 32.2 |
| | 9-16 | 117 | 43.8 |
| | 17-23 | 64 | 24 |

According to participants, the majority of them use smartphone (86.1%), installed Android operating system (82%), do not use mHealth apps for patient care (70.8%), and believed that mHealth can be useful in nursing care on average to a very high degree (85%). The most common times of using health-related apps were three to five times a day (36.7%), and friends were the most frequent factor in choosing the mobile apps used (40.1%) (Table 2).

The highest and lowest levels of awareness among nurses regarding the capabilities of mHealth were related to "transferring data from mHealth to information systems" (3.92 ± 0.912) and "creating a reminder to record data for the patient" (3.28 ± 1.029), respectively. Furthermore, the highest and lowest levels of attitude among nurses were related to "improvement of nursing work effectiveness" (3.82 ± 0.772) and "mHealth can be seen in the hospital where I work" (3.10 ± 1.162) (Table 3).

Almost 80% of the nurses stated that their satisfaction regarding their awareness about the concept of mHealth was at an average or lower level. About 92.9 percent of the participants, declare that didn't have any education relating to mHealth. Furthermore, nurses expressed their greatest willingness to participate in training courses related to mHealth (98.1%), electronic health records (96.3%), telemedicine (95.9%), electronic health (95.5%), and privacy, confidentiality, and health information security (93.6%) declared, respectively.

The low level of DHL of patients (4.19 ± 0.978), the reluctance of patients to use mHealth apps (4.03 ± 0.951), and the lack of nurses' awareness regarding desirable mHealth apps to recommend their use (3.70 ± 0.973) have been mentioned as the most important obstacles to the use of mHealth by nurses (Table 4).

The mean scores of nurses' awareness and attitude were evaluated as a relatively desirable.

Pearson's correlation coefficient showed a significant fair correlation between the nurses' awareness and attitude toward mHealth capabilities (r = 0. 289, P < 0.05). There was a significant poor correlation between nurses' awareness and obstacles to using mHealth (r = 0. 171, P < 0.05) and participants' educational level (r = 0. 119, P < 0.05). A significant poor correlation was observed between nurses' attitude and their educational level (r = 0.121, P < 0.05). Furthermore, a significant strong correlation was between participants' age and work experience (r = 0.943, P < 0.05). Additionally, there was a significant fair negative correlation between sex and age (r = - 0.333, P < 0.05) and work experience (r = -0.429, P < 0.05) (Table 5).

**Table 2. Frequency distribution of nurses' responses towards the mHealth.**

| Variables | Categories | Frequency | |
|---|---|---|---|
| | | **Number** | **Percent** |
| 1. Which of the mobile devices mostly do you use? | Smartphone | 230 | **86.1** |
| | Tablet | 30 | 11.2 |
| | Simple phone | 7 | 2.6 |
| 2. If you use a smartphone or tablet, which of the following is the operating system of your device? | Android | 219 | **82** |
| | iOS | 48 | 18 |
| 3. Which of the following health-related applications do you use? | Fitness | 89 | **33.3** |
| | Women's pregnancy and health | 35 | 13.1 |
| | Stress and lifestyle | 34 | 12.7 |
| | Nutrition and diet | 30 | 11.2 |
| | Alert and reminders | 30 | 11.2 |
| | Special disease | 22 | 8.2 |
| | I don't use | 27 | 10.1 |
| 4. How often do you use these health-related apps? | Every few days a weak | 57 | 21.3 |
| | Once a day | 96 | 36 |
| | Two to five times a day | 98 | **36.7** |
| | Six to 10 times a day | 16 | 6 |
| | 11 or more times a day | 0 | 0 |
| 5. Which of the mentioned factors played a role in your choice of health application? | Friends | 107 | **40.1** |
| | Internet | 78 | 29.2 |
| | Family | 56 | 21 |
| | Medical staff | 13 | 4.9 |
| | Medical journals | 6 | 2.2 |
| | App Store and similar stores | 5 | 1.9 |
| | TV/Radio | 2 | 0.7 |
| 6. Do you use mHealth apps for patient care? | Yes | 78 | 29.2 |
| | No | 189 | **70.8** |
| 7. In your opinion, how useful can mHealth apps be in nursing care? | Very high | 31 | 11.6 |
| | High | 89 | 33.3 |
| | Average | 107 | **40.1** |
| | Low | 34 | 12.8 |
| | Very low | 6 | 2.2 |

## Discussion

In this study, awareness, attitude, and obstacles the use of mHealth were evaluated from the nurses' perspective. The majority of nurses use smartphones (86%) and Android operating systems (82%). Fitness, women's pregnancy and health, and stress and lifestyle were the most frequent health related-apps used by them. Once a day (36%) and two to five times a week (36%) were the most frequent among the number of times nurses used health-related programs. Nurses have identified friends (40%), the internet (29%), and family (21%) as the most influential factors in choosing health-related applications. Most of the participants believed that mHealth programs can be beneficial in nursing care (85%).

In the present study, about one-third of nurses (29%) declared that they use mHealth apps for patient care. In Iran, Rahimi et al. [28] showed that about 37% of nurses use mHealth apps in their professional tasks. Our result was similar to the previous study conducted in Iran but

**Table 3. Awareness and attitude of nurses towards the capabilities of mHealth apps.**

| Dimensions and related questions | Likert scale N (%) | | | | | Mean ± S.D. |
|---|---|---|---|---|---|---|
| **Awareness** | Not at all aware | Slightly aware | Somewhat aware | Moderately aware | Extremely aware | |
| 1. Providing information in different formats (image, video, text) for users | 9 (3.4) | 10 (3.7) | 64 (24) | 130 (48.7) | 54 (20.2) | 3.79 ± 0.802 |
| 2. Automatic data recording | 5 (1.9) | 17 (6.4) | 53 (19.8) | 126 (47.2) | 66 (24.7) | 3.87 ± 0.924 |
| 3. Automatic data transfer to the Hospital Information System (HIS) or Electronic Health Record (EHR) | 4 (1.5) | 18 (6.7) | 44 (16.5) | 130 (48.7) | 71 (26.6) | **3.92 ± 0.912** |
| 4. Reminding the patient to enter data | 12 (4.5) | 52 (19.5) | 79 (29.6) | 98 (36.7) | 26 (9.7) | 3.28 ± 1.029 |
| 5. Reminding the patient to take medicine | 7 (2.6) | 18 (6.8) | 94 (35.2) | 93 (34.8) | 55 (20.6) | 3.64 ± 0.968 |
| 6. Display the patient's treatment plan | 5 (1.9) | 10 (3.7) | 62 (23.2) | 121 (45.3) | 69 (25.8) | 3.90 ± 0.895 |
| 7. Graphical display of data | 4 (1.5) | 5 (1.9) | 68 (25.5) | 123 (46.1) | 67 (25.1) | 3.91 ± 0.843 |
| 8. Providing communication between medical staff and patients | 14 (5.2) | 19 (7.1) | 76 (28.5) | 107 (40.1) | 51 (19.1) | 3.61 ± 1.040 |
| **Attitude** | Completely disagree | Disagree | Undecided | Agree | Completely agree | Mean ± S.D. |
| 1. mHealth accelerates the performance of my nursing duties. | 2 (0.7) | 24 (9) | 73 (27.3) | 144 (53.9) | 24 (9) | 3.61 ± 0.802 |
| 2. mHealth improves the quality of my nursing work. | 1 (0.4) | 17 (6.4) | 56 (21) | 163 (61) | 30 (11.2) | 3.76 ± 0.746 |
| 3. mHealth creates more security for providing health services. | 10 (3.7) | 31 (11.6) | 67 (25.1) | 132 (49.4) | 27 (10.1) | 3.51 ± 0.955 |
| 4. mHealth facilitates my nursing work. | 2 (0.7) | 29 (10.9) | 44 (16.5) | 157 (58.8) | 35 (13.1) | 3.73 ± 0.852 |
| 5. mHealth improves my work performance. | 2 (0.7) | 20 (7.5) | 63 (23.6) | 145 (54.3) | 37 (13.9) | 3.73 ± 0.819 |
| 6. mHealth improves the effectiveness of my work. | 3 (1.1) | 10 (3.7) | 59 (22.1) | 154 (57.7) | 41 (15.4) | **3.82 ± 0.772** |
| 7. mHealth gives me more control over my work. | 1 (0.4) | 29 (10.9) | 68 (25.5) | 134 (50.2) | 35 (13.1) | 3.65 ± 0.856 |
| 8. mHealth is perfectly compatible with my current position. | 26 (9.7) | 82 (30.7) | 118 (44.2) | 41 (15.4) | 0 (0) | 3.65 ± 0.855 |
| 9. mHealth in health care is simple for me. | 3 (1.1) | 26 (9.7) | 78 (29.2) | 121 (45.3) | 39 (14.6) | 3.63 ± 0.890 |
| 10. In this hospital, I see that mHealth is used to perform many tasks. | 32 (12) | 40 (15) | 59 (22.1) | 114 (42.7) | 22 (8.2) | 3.20 ± 1.162 |
| 11. Sharing medical information is simple using mHealth. | 6 (2.2) | 13 (4.9) | 69 (25.8) | 153 (57.3) | 26 (9.7) | 3.67 ± 0.806 |
| 12. mHealth can be seen in the hospital where I work. | 38 (14.2) | 40 (15) | 61 (22.8) | 114 (42.7) | 14 (5.2) | 3.10 ± 1.162 |
| 13. Using mHealth requires a lot of mental effort. | 16 (6) | 66 (24.7) | 73 (27.3) | 95 (35.6) | 17 (6.4) | 3.12 ± 1.043 |
| 14. Using mHealth is often frustrating. | 9 (3.4) | 57 (21.3) | 84 (31.5) | 99 (37.1) | 18 (6.7) | 3.22 ± 0.970 |
| 15. I recommend the use of mHealth in the nursing profession. | 12 (4.5) | 21 (7.9) | 56 (21) | 142 (53.2) | 36 (13.5) | 3.63 ± 0.966 |

**Table 4. Frequency distribution of obstacles to using mHealth from the nurses' perspectives.**

| Dimensions and related questions | Likert scale N (%) | | | | | Mean ± S.D. |
|---|---|---|---|---|---|---|
| | Very low | Below Average | Average | Above Average | Very High | |
| 1. Lack of access to trusted smart devices | 17 (6.4) | 34 (12.7) | 89 (33.3) | 78 (29.2) | 49 (18.4) | 3.40 ± 1.118 |
| 2. Lack of infrastructure (such as lack of Wi-Fi access) | 21 (7.9) | 19 (7.1) | 78 (29.2) | 89 (33.3) | 60 (22.5) | 3.55 ± 1.147 |
| 3. Lack of nurses' awareness about desirable apps to recommend their use | 5 (1.9) | 23 (8.6) | 79 (29.6) | 100 (37.5) | 60 (22.5) | 3.70 ± 0.973 |
| 4. Lack of time to discuss about apps during nursing care | 12 (4.5) | 38 (14.2) | 106 (39.7) | 88 (33) | 23 (8.6) | 3.27 ± 0.850 |
| 5. Failure to cover patient needs by apps | 14 (5.2) | 33 (12.4) | 85 (31.8) | 87 (32.6) | 48 (18) | 3.46 ± 1.083 |
| 6. The cost of applications | 42 (15.7) | 32 (12) | 79 (29.6) | 78 (29.2) | 36 (13.5) | 3.13 ± 1.253 |
| 7. The difficulty of using apps (lack of applicability) | 15 (5.6) | 49 (18.4) | 70 (26.2) | 93 (34.8) | 40 (15) | 3.35 ± 1.112 |
| 8. Patients' reluctance to use mHealth apps | 2 (0.7) | 15 (5.6) | 60 (22.5) | 87 (32.6) | 103 (38.6) | **4.03 ± 0.951** |
| 9. Lack of necessary efficiency in apps for nursing care. | 27 (10.1) | 31 (11.6) | 103 (38.6) | 84 (31.5) | 22 (8.2) | 3.16 ± 1.850 |
| 10. The low level of digital health literacy (DHL) of patients to use mHealth | 6 (2.2) | 12 (4.5) | 35 (13.1) | 87 (32.6) | 127 (47.6) | **4.19 ± 0.978** |
| 11. The low level of DHL of nurses to use mHealth | 27 (10.1) | 36 (13.5) | 78 (29.2) | 91 (34.1) | 35 (13.1) | 3.27 ± 1.157 |
| 12. Uncertainty of data security in mHealth platform | 2 (0.7) | 30 (11.2) | 87 (32.6) | 120 (44.9) | 28 (10.5) | 3.53 ± 0.855 |
| 13. Uncertainty about the software (application) used | 3 (1.1) | 36 (13.5) | 79 (29.6) | 116 (43.4) | 33 (12.4) | 3.52 ± 0.915 |
| 14. Uncertainty about the device used (mobile phone, tablet, etc.) | 8 (3) | 29 (10.9) | 96 (36) | 102 (38.2) | 32 (12) | 3.45 ± 0.942 |

**Table 5. Correlation between evaluated variables.**

| Variables | | Mean ± S.D. | 1 | 2 | 3 | 4 | 5 | 6 | 7 |
|---|---|---|---|---|---|---|---|---|---|
| 1 | Awareness | 3.74 ± 0.657 | 1 | | | | | | |
| 2 | Attitude | 3.49 ± 0.513 | 0.289* | 1 | | | | | |
| 3 | Obstacles | 3.50 ± 0.597 | 0.171* | -0.031 | 1 | | | | |
| 4 | Age | 35.74 ± 5.326 | 0.070 | -0.065 | -0.008 | 1 | | | |
| 5 | Work Experience | 11.61 ± 5.487 | 0.059 | -0.058 | 0.015 | 0.943* | 1 | | |
| 6 | Education level | – | 0.119* | 0.121* | 0.006 | 0.011 | 0.004 | 1 | |
| 7 | Sex | – | -0.020 | -0.089 | -0.007 | -0.333* | -0.429* | 0.012 | 1 |

* P- value < 0.05 were considered statistically significant.

contradicted with Mayer et al. study results [23]. This contradiction could be explained by different knowledge levels of nurses regarding mHealth apps.

The effective use of mHealth apps by healthcare practitioners requires that they have sufficient knowledge about these apps [36,37]. Our results showed that the mean score of nurses' awareness regarding mHealth apps was 3.74 ± 0.657 out of 5, indicating that nurses have a relatively desirable level of awareness in this regard. Raj [30] reported that the mean score of nurses' knowledge about mHealth was 6.47, 7.79, and 4.64 out of a total of 10 in total, in Finland, and Lithuania respectively. In India, Elizebeth and Christena [38] specified that only 30% and 22% of nurses had average and good knowledge of mHealth, respectively. In Nigeria, Owolabi et al. [39] determined that about one-third of nurses are aware of mHealth services. Our finding was in line with [30,38] but contradicted the findings of Owolabi et al. [39]. The available contradiction could be justified by the difference in the level of digital health literacy (DHL) of the participants and the training provided to them in these studies.

The willingness to adopt and use mHealth effectively is largely determined by consumer attitude, as shown by previous studies [29,40]. In addition, the positive attitude towards mHealth is influenced by the various benefits expected by users from this technology [41]. The mean score of nurses' attitudes concerning mHealth apps was 3.49 ± 0.513 out of 5, which showed a relatively desired level. Raj [30] conducted a study in Finland and Lithuania and reported a mean score of 3.48 out of 5 about nurses' attitudes toward mHealth. In South Africa, Lekalakala-Mokgele et al. [42] indicated that the mean score of nursing students' attitudes toward eHealth was 3.03 ± 0.73. Increasing users' awareness of the capabilities and potential benefits of using mHealth is a key solution to improve their attitude towards this technology [43]. Considering the desire of the majority of nurses participating in the study (98%) to participate in the training courses of mHealth, it is possible to improve the attitude of nurses regarding the use of this technology.

The average score of 3.50 ± 0.597 out of 5 was obtained regarding the obstacles to using mHealth from the nurses' perspective, which indicates the presence of relatively high barriers in the practical use of this technology by nurses. Our findings revealed that the low level of DHL of patients to use mHealth, patients' reluctance to use mHealth apps, and lack of nurses' awareness about desirable apps to recommend its use for patients as the most important obstacles to the use of mHealth apps. DHL can be described as the capability to effectively seek, discover, comprehend, and assess health-related information from digital sources, and subsequently apply the acquired knowledge to address health-associated concerns [44,45].

In the UK, Ali et al. [32] reported increasing costs and advanced patient age as the main barriers to the adoption of mHealth by healthcare professionals. In the United States, Zhou et al. [46] reported the cost of apps and security issues as the main barriers to nurses' use of

mHealth in patient care. In South Korea, Boo and Oh [47], technical challenges and inequality of patients' level of DHL were reported as the main obstacles to using mHealth from the nurses' perspective. Thus, improving the DHL of health care professionals and especially patients, free access to mHealth apps, guaranteeing the security and simplicity of using these apps, and training users to improve their understanding of the usefulness and willingness to use mobile apps to remove the existing barriers is essential.

There was a significant positive correlation between nurses' awareness with their attitude about mHealth and obstacles, which indicates that higher awareness of nurses about mHealth is directly related to a more positive attitude about using it and a better understanding of the obstacles in using this technology. Furthermore, a significant negative correlation was observed between nurses' attitudes and obstacles to using mHealth. This relationship indicates that the positive attitude of nurses towards the use of mHealth can make them feel less obstacles in using this technology.

## Strengths and limitation

In this study, clear yet rigorous statistical analysis, following sound research methodology, has been used to provide scientific insight into the awareness, attitude, and obstacles to using mHealth by nurses, which provides the possibility of conducting research in similar contexts. Therefore, policy-makers and managers can use the findings to make the necessary policy and planning to use mHealth technology to provide health care services to patients by nurses.

There are several limitations to this study. Participants were recruited from public hospitals, while participants from other contexts such as private hospitals were not included. Thus, the generalizability of the findings to other settings may be limited. The study may have faced self-reporting biases due to the use of a self-reported and cross-sectional survey. This study was conducted in a city in the southeast of Iran, and therefore some of our results cannot be generalized to the whole country.

## Conclusion

Most nurses in teaching hospitals affiliated with Zahedan University of Medical Sciences use smartphones, predominantly with Android operating systems, and recognize the benefits of mHealth apps in nursing care. Fitness, women's pregnancy and health, and stress and lifestyle were the most frequent mHealth apps they used. Still, obstacles like low DHL among patients, patient hesitation, and limited awareness of suitable apps impede nurses' practical implementation of mHealth. Improvement of the users' digital health literacy (nurses and especially patients), providing users with free access to mHealth apps, ensuring security and facilitating the usability of the apps, educating users, promoting the understanding of nurses about the benefits of mHealth and their willingness to use the technology is essential. Our findings provide an insight into the awareness and attitude of nurses towards mHealth and the obstacles to its use to provide a basis for policy-making and planning to increase the use of this technology. It is suggested that future studies be carried out to identify solutions to remove existing obstacles.

## Supporting information

**S1 Checklist.**
(DOCX)

**S1 File.**
(PDF)

## Acknowledgments

The authors thank the nurses who participated in this study for sharing their valuable experiences.

## Author contributions

**Conceptualization:** Jahanpour Alipour, Somayyeh Zakerabasali, Afsaneh Karimi.

**Data curation:** Jahanpour Alipour, Somayyeh Zakerabasali, Afsaneh Karimi.

**Formal analysis:** Jahanpour Alipour, Yousef Mehdipour, Somayyeh Zakerabasali, Afsaneh Karimi.

**Funding acquisition:** Jahanpour Alipour.

**Methodology:** Yousef Mehdipour.

**Writing – original draft:** Jahanpour Alipour, Yousef Mehdipour, Somayyeh Zakerabasali, Afsaneh Karimi.

**Writing – review & editing:** Jahanpour Alipour, Yousef Mehdipour, Somayyeh Zakerabasali, Afsaneh Karimi.

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
