## [Decision Letter · Decision Letter 0]

6 Nov 2024

PONE-D-24-31629Nurses' Perspectives on Using mHealth Apps in a Developing Country: Awareness, Attitude, and ObstaclesPLOS ONE

Dear Dr. Alipour,

Thank you for submitting your manuscript to PLOS ONE. After careful consideration, we feel that it has merit but does not fully meet PLOS ONE’s publication criteria as it currently stands. Therefore, we invite you to submit a revised version of the manuscript that addresses the points raised during the review process.

**The paper touched on very important area of digital health especially at this time when technology is driving most of initiatives in most fields including healthcare. Please review and address the following comments thoroughly including the comments from the reviewer 3 below.**

**Financial disclosure:**

**Please indicate whether the sponsors or funders play any role in the study design, data collection and analysis, decision to publish, or preparation of the manuscript?**

**Ethical approval:**

**Please add ethical approval as supporting document and remove the URL from methods section.**

**Methods:**

**Study population and sample: please provide inclusion/exclusion criteria for the participants****Data collection tool was explained in detail. However, data collection process was not explained. Please provide explanations about how the data was collected (paper-base or electronically), who collected the data (interviewer or self-administered), how was the data transmitted, stored and cleaned e.t.c.**

**Results:**

**Age group: please what was the basis for this categorization. The age intervals are not consistent across the age groups. Why and what is the rational for this. If there is any justification, please explain in the methods section otherwise use conventional age categorization with regular intervals.**

**Discussion:**

**Study limitations were provided. Any strengths for the study?**

We look forward to receiving your revised manuscript.

Kind regards,

Ibrahim Jahun, MD, MSc, PhD

Academic Editor

PLOS ONE

**Journal Requirements:**

This study was supported by the Vice Chancellor for Research and Technology of Zahedan University of Medical Sciences (No: 10494).

The authors thank the Vice Chancellor for Research and Technology of Zahedan University of Medical Sciences for supporting the project. They also thankful to the nurses who participated in this study for sharing their valuable experiences.

This study was supported by the Vice Chancellor for Research and Technology of Zahedan University of Medical Sciences (No: 10494).

6. Please amend either the abstract on the online submission form (via Edit Submission) or the abstract in the manuscript so that they are identical.

7. Your ethics statement should only appear in the Methods section of your manuscript. If your ethics statement is written in any section besides the Methods, please delete it from any other section. 

Reviewers' comments:

Reviewer's Responses to Questions

**Comments to the Author**

1. Is the manuscript technically sound, and do the data support the conclusions?

Reviewer #1: Yes

Reviewer #2: Partly

Reviewer #3: Yes

2. Has the statistical analysis been performed appropriately and rigorously? 

Reviewer #1: Yes

Reviewer #2: Yes

Reviewer #3: Yes

3. Have the authors made all data underlying the findings in their manuscript fully available?

Reviewer #1: Yes

Reviewer #2: Yes

Reviewer #3: Yes

4. Is the manuscript presented in an intelligible fashion and written in standard English?

Reviewer #1: Yes

Reviewer #2: Yes

Reviewer #3: Yes

5. Review Comments to the Author

**Reviewer #1:**  I consider the abstract to be technically sound and even more as a Nigerian, living in this time of coming of age of technology and given recent boom in mHealth technologies in developing nations, I consider it even more timely and apt. As a leader in the health informatics field, developing some of the mHealth innovations, I often stop to consider the impact of the many mHealth tools and solutions we churn out every now and then, targeting either the clinician or the patient. My understanding is that the author who I have worked with in the past, have decided to do more fundamental study to figure out this question around 'impact' and 'effectiveness'.

The studies have shown with clear and yet rigorous statistical analysis, following sound research methodology, what the answer to the question is by identifying critical variables of awareness, attitude, and obstacles and measuring their mean scores. Which anyone can attempt to reproduce if they wish. The Manuscript is easy to understand in first reading because of the authors have deployed intelligible straight-forward English narratives in their writing.

**Reviewer #2: ** The manuscript raises important problem of mobile applications suitability for nurses and medical workers

Overall, information is too limited. It is restricted to the study at one city. It should be generalized to whole region, to developing countries.

The presentation material itself is too limited. The manuscript has only tables, no figures.

One large figure – scheme of experiment – will improve the material presentation

In the Abstract:

mHealth Apps – should be commented. It is any mobile application, in general, or program by some manufacturer?

“Android operating system installed (82%)”

Don’t use passive voice in English (“were used”, “was done”). Write directly.

Results section shows rather common statistics. Conclusion is not complete.

The term “the digital health literacy” should be commented. It is kind of science jargon,

“Mobile Health”, “Obstacle” are not appropriate as keywords.

In the text “The estimate made in 2018…” - old data.

The term ‘mHealth’ refers to the paper in Persian. Need cite available literature, add references.

Please not use bulk citations – 3 or more references together, like “(15, 18-20)”, “(21-23)”. Separate phrases into parts, add detail.

Tables 3 and 4 – the results should be highlighted, show most interesting result, discuss it. Maybe present it in other visual form (like a histogram, heatmap).

Conclusion should have general interest. Phrase like “The majority of nurses use smartphones (86%) and Android operating systems (82%)” gives no information. Nurses where?

**Reviewer #3: ** Some Concerns:

1. The authors indicated in the limitation section that the study was conducted in one city in the southeast of Iran therefore the results cannot be generalized to the whole country not to talk of several developing countries. So, the geographical coverage, sample size and the findings in this paper does not aligning with title of the paper. Therefore, the author should consider narrowing the title to reflect the locations/region where the study was conducted. Example “Nurses' Perspectives on Using mHealth Apps in Southeastern Iran: Awareness, Attitude, and Obstacles”.

2. The purpose of the discussion section is to interpret results and justify conclusion. The discussion section seems to be too long, citing lots of works done in other countries, I suggest that some of those literatures should be moved to introduction section where past related works were quoted. You need to restate your key results in this section.

6. PLOS authors have the option to publish the peer review history of their article (what does this mean? ). If published, this will include your full peer review and any attached files.

**Do you want your identity to be public for this peer review?** For information about this choice, including consent withdrawal, please see our Privacy Policy .

Reviewer #1: **Yes: ** Emeka Christian Madubuko

Reviewer #2: No

Reviewer #3: **Yes: ** Mukhtar Liman Ahmed

---

## [Author Response · Author response to Decision Letter 1]

29 Nov 2024

Dear Academic Editor and Reviewers,

Thank you very much for your consideration, and we really appreciate the comments and have learned a lot. Appropriate changes were made in the revised manuscript according to the suggestions of reviewers and editor.

Responses to academic editor comments:

Financial disclosure:

• Please indicate whether the sponsors or funders play any role in the study design, data collection and analysis, decision to publish, or preparation of the manuscript?

• Answer: We appreciate your scientific and constructive comments. It was done accordingly in the financial disclosure section.

Ethical approval:

• Please add ethical approval as supporting document and remove the URL from methods section.

• Answer: It was done accordingly.

Methods:

• Study population and sample: please provide inclusion/exclusion criteria for the participants

• Answer: Inclusion criteria for participants were added in the method section. Page 5, lines 135-7

• Data collection tool was explained in detail. However, data collection process was not explained. Please provide explanations about how the data was collected (paper-base or electronically), who collected the data (interviewer or self-administered), how was the data transmitted, stored and cleaned e.t.c.

• Answer: data collection process was added accordingly. Page 5, lines 139-156

Results:

• Age group: please what was the basis for this categorization. The age intervals are not consistent across the age groups. Why and what is the rational for this. If there is any justification, please explain in the methods section otherwise use conventional age categorization with regular intervals.

• Answer: Despite the change and modification of the age categories and the recalculation of the relevant descriptive statistics (number and percentage) remained constant. Page 7, table 1.

Discussion:

• Study limitations were provided. Any strengths for the study?

• Answer: Strengths of the study were added. Page 12, lines 289-293.

Journal Requirements:

Answer: We reviewed the manuscript and made the necessary corrections.

This study was supported by the Vice Chancellor for Research and Technology of Zahedan University of Medical Sciences (No: 10494).

Answer: Thank you.

Answer: Done accordingly.

The authors thank the Vice Chancellor for Research and Technology of Zahedan University of Medical Sciences for supporting the project. They also thankful to the nurses who participated in this study for sharing their valuable experiences.

This study was supported by the Vice Chancellor for Research and Technology of Zahedan University of Medical Sciences (No: 10494).

Answer: Done accordingly.

Answer: We apologize for the unwanted ambiguity. We uploaded Supporting information files entitled “Raw data”.

Answer: Thanks for your advice. We have corrected this section to adhere to your open data policy.

6. Please amend either the abstract on the online submission form (via Edit Submission) or the abstract in the manuscript so that they are identical.

Answer: Done accordingly.

7. Your ethics statement should only appear in the Methods section of your manuscript. If your ethics statement is written in any section besides the Methods, please delete it from any other section.

Answer: Done accordingly. Page 5, lines 126-9.

Answer: Based on our review, the references were not problem.

Responses to Reviewers’ comments

Reviewer #1: I consider the abstract to be technically sound and even more as a Nigerian, living in this time of coming of age of technology and given recent boom in mHealth technologies in developing nations, I consider it even more timely and apt. As a leader in the health informatics field, developing some of the mHealth innovations, I often stop to consider the impact of the many mHealth tools and solutions we churn out every now and then, targeting either the clinician or the patient. My understanding is that the author who I have worked with in the past, have decided to do more fundamental study to figure out this question around 'impact' and 'effectiveness'.

The studies have shown with clear and yet rigorous statistical analysis, following sound research methodology, what the answer to the question is by identifying critical variables of awareness, attitude, and obstacles and measuring their mean scores. Which anyone can attempt to reproduce if they wish. The Manuscript is easy to understand in first reading because of the authors have deployed intelligible straight-forward English narratives in their writing.

Answer: We are grateful to the reviewer#1 for your appreciation of the value of our work.

Reviewer #2: The manuscript raises important problem of mobile applications suitability for nurses and medical workers

Overall, information is too limited. It is restricted to the study at one city. It should be generalized to whole region, to developing countries.

The presentation material itself is too limited. The manuscript has only tables, no figures.

One large figure – scheme of experiment – will improve the material presentation

Answer: We appreciate the reviewer' valuable comments and constructive suggestions, which help improve the quality of the manuscript.

In the Abstract:

mHealth Apps – should be commented. It is any mobile application, in general, or program by some manufacturer? “Android operating system installed (82%)”

Answer: We are sorry for the unwanted ambiguity. We meant it in general. But the results of the study showed that most nurses use the Android operating system on their devices. We modified the title of the manuscript as well as the introduction subsection of the abstract section for clarity. Page 2, lines 33-35.

Don’t use passive voice in English (“were used”, “was done”). Write directly.

Answer: We appreciate your scientific comment and suggestion. We revised the sentences in the abstract and methods sections and used active voice sentences instead of passive ones.

Results section shows rather common statistics. Conclusion is not complete.

Answer: We agree with the reviewer. We revised the conclusion section to clear the ambiguity. Page 2, lines 49-56.

“Mobile Health”, “Obstacle” are not appropriate as keywords.

Answer: Both keywords “Mobile Health”and mHealth have been provided as entry terms of telemedicine the search for the keyword mHealth in Mesh Browser. Therefore, we added telemedicine to keywords. In addition, we removed the obstacle from the keywords section. If the reviewer does not agree with the keywords provided by us, we would be grateful if the reviewer would guide us in determining the keywords. Page 2, line 59.

The term “the digital health literacy” should be commented. It is kind of science jargon,

Answer: We agree with the reviewer. For this reason, we added a scientific and comprehensive definition in the discussion section to clear the ambiguity. Page 12, lines 269-72.

In the text “The estimate made in 2018…” - old data.

Answer: Thank you very much for the completely scientific comment of the reviewer. The indicated sentence was amended with a new sentence and date. Page 3, lines 68-9.

The term ‘mHealth’ refers to the paper in Persian. Need cite available literature, add references.

Answer: We agree with the reviewer and we added related references accordingly. Page 3, lines 70-72.

Please not use bulk citations – 3 or more references together, like “(15, 18-20)”, “(21-23)”. Separate phrases into parts, add detail.

Answer: We agree with the reviewer. we corrected all the cases that had conditions similar to those mentioned by the reviewer throughout the manuscript. Pages 3 and 4, lines 64-110.

Tables 3 and 4 – the results should be highlighted, show most interesting result, discuss it. Maybe present it in other visual form (like a histogram, heatmap).

Answer: According to the reviewer suggestions, most interesting result highlighted in the tables.

We agree with the reviewer that the results are better in the form of a combination of tables and visual format (especially in graphs form) in the articles. However, in our article, Tables 3 and 4 contain the relevant details (Likert scale) and present the mean scores along with the relevant standard deviation, which cannot be expressed if the data is presented using visual format. Therefore, with respect to the reviewer's comment, we believe that presenting the data in the current format is better.

Conclusion should have general interest. Phrase like “The majority of nurses use smartphones (86%) and Android operating systems (82%)” gives no information. Nurses where?

Answer: we agree with the reviewer and apologize for any unintended ambiguity. we have revised the conclusion section to enhance clarity. Page 13, lines 301-12.

Reviewer #3: Some Concerns:

1. The authors indicated in the limitation section that the study was conducted in one city in the southeast of Iran therefore the results cannot be generalized to the whole country not to talk of several developing countries. So, the geographical coverage, sample size and the findings in this paper does not aligning with title of the paper. Therefore, the author should consider narrowing the title to reflect the locations/region where the study was conducted. Example “Nurses' Perspectives on Using mHealth Apps in Southeastern Iran: Awareness, Attitude, and Obstacles”.

Answer: Thank you for the constructive comments to our manuscript. We agree with the reviewer comment and suggestion. We modified the title accordingly as follows. Page 1, lines 1-2.

"Nurses' perspectives on using mHealth apps in southeastern Iran: Awareness, attitude, and obstacles"

2. The purpose of the discussion section is to interpret results and justify conclusion. The discussion section seems to be too long, citing lots of works done in other countries, I suggest that some of those literatures should be moved to introduction section where past related works were quoted. You need to restate your key results in this section.

Answer: We agree with the reviewer suggestion. We moved two literatures from the discussion to the introduction section. In addition, we removed two less relevant literature items from the discussion section. Page 4, lines 109-11.

---

## [Decision Letter · Decision Letter 1]

16 Dec 2024

Nurses' perspectives on using mobile health applications in southeastern Iran: Awareness, attitude, and obstacles

PONE-D-24-31629R1

Dear Dr. Alipour,

We’re pleased to inform you that your manuscript has been judged scientifically suitable for publication and will be formally accepted for publication once it meets all outstanding technical requirements.

Kind regards,

Ibrahim Jahun, MD, MSC, PhD

Academic Editor

PLOS ONE

Additional Editor Comments (optional):

Reviewers' comments:

Reviewer's Responses to Questions

**Comments to the Author**

1. If the authors have adequately addressed your comments raised in a previous round of review and you feel that this manuscript is now acceptable for publication, you may indicate that here to bypass the “Comments to the Author” section, enter your conflict of interest statement in the “Confidential to Editor” section, and submit your "Accept" recommendation.

Reviewer #3: All comments have been addressed

2. Is the manuscript technically sound, and do the data support the conclusions?

Reviewer #3: Yes

3. Has the statistical analysis been performed appropriately and rigorously? 

Reviewer #3: Yes

4. Have the authors made all data underlying the findings in their manuscript fully available?

Reviewer #3: Yes

5. Is the manuscript presented in an intelligible fashion and written in standard English?

Reviewer #3: Yes

6. Review Comments to the Author

Reviewer #3: The authors have reflected my review comments in the revised copy of the paper. I have no further comments.

7. PLOS authors have the option to publish the peer review history of their article (what does this mean? ). If published, this will include your full peer review and any attached files.

**Do you want your identity to be public for this peer review?** For information about this choice, including consent withdrawal, please see our Privacy Policy .

Reviewer #3: **Yes: ** Mukhtar Liman Ahmed

---

## [Editor Report · Acceptance letter]

PONE-D-24-31629R1

PLOS ONE

Dear Dr. Alipour,

I'm pleased to inform you that your manuscript has been deemed suitable for publication in PLOS ONE. Congratulations! Your manuscript is now being handed over to our production team.

Kind regards,

on behalf of

Dr. Ibrahim Jahun

Academic Editor

PLOS ONE